# Scalar Posterior Sampling with Applications

**Georgios Theocharous**
Adobe Research
theochar@adobe.com

**Zheng Wen**
Adobe Research
zwen@adobe.com

**Yasin Abbasi-Yadkori**
Adobe Research
abbasiya@adobe.com

**Nikos Vlassis**
Netflix
nvlassis@netflix.com

## Abstract

We propose a practical non-episodic PSRL algorithm that unlike recent state-of-the-art PSRL algorithms uses a deterministic, model-independent episode switching schedule. Our algorithm termed deterministic schedule PSRL (DS-PSRL) is efficient in terms of time, sample, and space complexity. We prove a Bayesian regret bound under mild assumptions. Our result is more generally applicable to multiple parameters and continuous state action problems. We compare our algorithm with state-of-the-art PSRL algorithms on standard discrete and continuous problems from the literature. Finally, we show how the assumptions of our algorithm satisfy a sensible parametrization for a large class of problems in sequential recommendations.

## 1 Introduction

Thompson sampling [Thompson, 1933], or posterior sampling for reinforcement learning (PSRL), is a conceptually simple approach to deal with unknown MDPs [Strens, 2000; Osband *et al.*, 2013]. PSRL begins with a prior distribution over the MDP model parameters (transitions and/or rewards) and typically works in episodes. At the start of each episode, an MDP model is sampled from the posterior belief and the agent follows the policy that is optimal for that sampled MDP until the end of the episode. The posterior is updated at the end of every episode based on the observed actions, states, and rewards. A special case of MDP under which PSRL has been recently extensively studied is MDP with state resetting, either explicitly or implicitly. Specifically, in [Osband *et al.*, 2013; Osband and Van Roy, 2014] the considered MDPs are assumed to have fixed-length episodes, and at the end of each episode the MDP's state is reset according to a fixed state distribution. In [Gopalan and Mannor, 2015], there is an assumption that the environment is ergodic and that there exists a recurrent state under any policy. Both approaches have developed variants of PSRL algorithms under their respective assumptions, as well as state-of-the-art regret bounds, Bayesian in [Osband *et al.*, 2013; Osband and Van Roy, 2014] and Frequentist in [Gopalan and Mannor, 2015].

However, many real-world problems are of a continuing and non-resetting nature. These include sequential recommendations and other common examples found in controlled mechanical systems (e.g., control of manufacturing robots), and process optimization (e.g., controlling a queuing system), where 'resets' are rare or unnatural. Many of these real world examples could easily be parametrized with a scalar parameter, where each value of the parameter could specify a complete model. These type of domains do not have the luxury of state resetting, and the agent needs to learn to act, without necessarily revisiting states. Extensions of the PSRL algorithms to general MDPs without state resetting has so far produced non-practical algorithms and in some cases buggy theoretical analysis. This is due to the difficulty of analyzing regret under policy switching schedules that depend on various dynamic statistics produced by the true underlying model (e.g., doubling the visitations of

state and action pairs and uncertainty reduction of the parameters). Next we summarize the literature for this general case PSRL.

The earliest such general case was analyzed as Bayes regret in a 'lazy' PSRL algorithm [Abbasi-Yadkori and Szepesvári, 2015]. In this approach a new model is sampled, and a new policy is computed from it, every time the uncertainty over the underlying model is sufficiently reduced; however, the corresponding analysis was shown to contain a gap [Osband and Van Roy, 2016].

A recent general case PSRL algorithm with Bayes regret analysis was proposed in [Ouyang *et al.*, 2017b]. At the beginning of each episode, the algorithm generates a sample from the posterior distribution over the unknown model parameters. It then follows the optimal stationary policy for the sampled model for the rest of the episode. The duration of each episode is dynamically determined by two stopping criteria. A new episode starts either when the length of the current episode exceeds the previous length by one, or when the number of visits to any state-action pair is doubled. They establish $\widetilde{O}(HS\sqrt{AT})$ bounds on expected regret under a Bayesian setting, where $S$ and $A$ are the sizes of the state and action spaces, $T$ is time, and $H$ is the bound of the span, and $\widetilde{O}$ notation hides logarithmic factors. However, despite the state-of-the-art regret analysis, the algorithm is not well suited for large and continuous state and action spaces due to the requirement to count state and action visitations for all state-action pairs.

In another recent work [Agrawal and Jia, 2017], the authors present a general case PSRL algorithm that achieves near-optimal worst-case regret bounds when the underlying Markov decision process is communicating with a finite, though unknown, diameter. Their main result is a high probability regret upper bound of $\widetilde{O}(D\sqrt{SAT})$ for any communicating MDP with $S$ states, $A$ actions and diameter $D$, when $T \geq S^5 A$. Despite the nice form of the regret bound, this algorithm suffers from similar practicality issues as the algorithm in [Ouyang *et al.*, 2017b]. The epochs are computed based on doubling the visitations of state and action pairs, which implies tabular representations. In addition it employs a stricter assumption than previous work of a fully communicating MDP with some unknown diameter. Finally, in order for the bound to be true $T \geq S^5 A$, which would be impractical for large scale problems.

Both of the above two recent state-of-the-art algorithms [Ouyang *et al.*, 2017b; Agrawal and Jia, 2017], do not use generalization, in that they learn separate parameters for each state-action pair. In such non-parametrized case, there are several other modern reinforcement learning algorithms, such as UCRL2 [Jaksch *et al.*, 2010], REGAL [Bartlett and Tewari, 2009], and R-max [Brafman and Tennenholtz, 2002], which learn MDPs using the well-known 'optimism under uncertainty' principle. In these approaches a confidence interval is maintained for each state-action pair, and observing a particular state transition and reward provides information for only that state and action. Such approaches are inefficient in cases where the whole structure of the MDP can be determined with a scalar parameter.

Despite the elegant regret bounds for the general case PSRL algorithms developed in [Ouyang *et al.*, 2017b; Agrawal and Jia, 2017], both of them focus on tabular reinforcement learning and hence are sample inefficient for many practical problems with exponentially large or even continuous state/action spaces. On the other hand, in many practical RL problems, the MDPs are parametrized in the sense that system dynamics and reward/loss functions are assumed to lie in a known parametrized low-dimensional manifold [Gopalan and Mannor, 2015]. Such model parametrization (i.e. model generalization) allows researchers to develop sample efficient algorithms for large-scale RL problems. Our paper belongs to this line of research. Specifically, we propose a novel general case PSRL algorithm, referred to as DS-PSRL, that exploits model parametrization (generalization). We prove an $\widetilde{O}(\sqrt{T})$ Bayes regret bound for DS-PSRL, assuming we can model every MDP with a single smooth parameter.

DS-PSRL also has lower computational and space complexities than algorithms proposed in [Ouyang *et al.*, 2017b; Agrawal and Jia, 2017]. In the case of [Ouyang *et al.*, 2017b] the number of policy switches in the first $T$ steps is $K_T = O\left(\sqrt{2SAT log(T)}\right)$; on the other hand, DS-PSRL adopts a deterministic schedule and its number of policy switches is $K_T \leq \log(T)$. Since the major computational burden of PSRL algorithms is to solve a sampled MDP at each policy switch, DS-PSRL is computationally more efficient than the algorithm proposed in [Ouyang *et al.*, 2017b]. As to the space complexity, both algorithms proposed in [Ouyang *et al.*, 2017b; Agrawal and Jia, 2017]

need to store counts of state and action visitations. In contrast, DS-PSRL uses a model independent schedule and as a result does not need to store such statistics.

In the rest of the paper we will describe the DS-PSRL algorithm, and derive a state-of-the-art Bayes regret analysis. We will demonstrate and compare our algorithm with state-of-the-art on standard problems from the literature. Finally, we will show how the assumptions of our algorithm satisfy a sensible parametrization for a large class of problems in sequential recommendations.

## 2   Problem Formulation

We consider the reinforcement learning problem in a parametrized Markov decision process (MDP) $(\mathcal{X}, \mathcal{A}, \ell, P^{\theta_*})$ where $\mathcal{X}$ is the state space, $\mathcal{A}$ is the action space, $\ell : \mathcal{X} \times \mathcal{A} \to \mathbb{R}$ is the instantaneous loss function, and $P^{\theta_*}$ is an MDP transition model parametrized by $\theta_*$. We assume that the learner knows $\mathcal{X}$, $\mathcal{A}$, $\ell$, and the mapping from the parameter $\theta_*$ to the transition model $P^{\theta_*}$, but does not know $\theta_*$. Instead, the learner has a prior belief $P_0$ on $\theta_*$ at time $t = 0$, before it starts to interact with the MDP. We also use $\Theta$ to denote the support of the prior belief $P_0$. Note that in this paper, we do not assume $\mathcal{X}$ or $\mathcal{A}$ to be finite; they can be infinite or even continuous. For any time $t = 1, 2, \ldots$, let $x_t \in \mathcal{X}$ be the state at time $t$ and $a_t \in \mathcal{A}$ be the action at time $t$. Our goal is to develop an algorithm (controller) that adaptively selects an action $a_t$ at every time step $t$ based on prior information and past observations to minimize the long-run Bayes average loss

$$\mathbb{E}\left[\limsup_{n\to\infty} \frac{1}{n} \sum_{t=1}^{n} \ell(x_t, a_t)\right].$$

Similarly as existing literature [Osband *et al.*, 2013; Ouyang *et al.*, 2017b], we measure the performance of such an algorithm using Bayes regret:

$$R_T = \mathbb{E}\left[\sum_{t=1}^{T} \left(\ell(x_t, a_t) - J_{\pi^*}^{\theta_*}\right)\right], \tag{1}$$

where $J_{\pi^*}^{\theta_*}$ is the average loss of running the optimal policy under the true model $\theta_*$. Note that under the mild 'weakly communicating' assumption, $J_{\pi^*}^{\theta_*}$ is independent of the initial state.

The Bayes regret analysis of PSRL relies on the key observation that at each stopping time $\tau$ the true MDP model $\theta_*$ and the sampled model $\widetilde{\theta}_\tau$ are identically distributed [Ouyang *et al.*, 2017b]. This fact allows to relate quantities that depend on the true, but unknown, MDP $\theta_*$, to those of the sampled MDP $\widetilde{\theta}_\tau$ that is fully observed by the agent. This is formalized by the following Lemma 1.

**Lemma 1** *(Posterior Sampling [Ouyang* et al.*, 2017b]). Let $(\mathcal{F}_s)_{s=1}^{\infty}$ be a filtration ($\mathcal{F}_s$ can be thought of as the historic information until current time $s$) and let $\tau$ be an almost surely finite $\mathcal{F}_s$-stopping time. Then, for any measurable function $g$,*

$$\mathbb{E}\left[g(\theta_*)|\mathcal{F}_\tau\right] = \mathbb{E}\left[g(\widetilde{\theta}_\tau)|\mathcal{F}_\tau\right]. \tag{2}$$

*Additionally, the above implies that $\mathbb{E}\left[g(\theta_*)\right] = \mathbb{E}\left[g(\widetilde{\theta}_\tau)\right]$ through the tower property.*

## 3   The Proposed Algorithm: Deterministic Schedule PSRL

In this section, we propose a PSRL algorithm with a deterministic policy update schedule, shown in Figure 1. The algorithm changes the policy in an exponentially rare fashion; if the length of the current episode is $L$, the next episode would be $2L$. This switching policy ensures that the total number of switches is $O(\log T)$. We also note that, when sampling a new parameter $\widetilde{\theta}_t$, the algorithm finds the optimal policy assuming that the sampled parameter is the true parameter of the system. Any planning algorithm can be used to compute this optimal policy [Sutton and Barto, 1998]. In our analysis, we assume that we have access to the exact optimal policy, although it can be shown that this computation need not be exact and a near optimal policy suffices (see [Abbasi-Yadkori and Szepesvári, 2015]).

```
Inputs: P₁, the prior distribution of θ∗.
L ← 1.
for t ← 1, 2, . . . do
    if t = L then
        Sample θ̃ₜ ∼ Pₜ.
        L ← 2L.
    else
        θ̃ₜ ← θ̃ₜ₋₁.
    end if
    Calculate near-optimal action aₜ ← π*(xₜ, θ̃ₜ).
    Execute action aₜ and observe the new state xₜ₊₁.
    Update Pₜ with (xₜ, aₜ, xₜ₊₁) to obtain Pₜ₊₁.
end for
```

Figure 1: The DS-PSRL algorithm with deterministic schedule of policy updates.

To measure the performance of our algorithm we use Bayes regret $R_T$ defined in Equation 1. The slower the regret grows, the closer is the performance of the learner to that of an optimal policy. If the growth rate of $R_T$ is sublinear ($R_T = o(T)$), the average loss per time step will converge to the optimal average loss as $T$ gets large, and in this sense we can say that the algorithm is asymptotically optimal. Our main result shows that, under certain conditions, the construction of such asymptotically optimal policies can be reduced to efficiently sampling from the posterior of $\theta_*$ and solving classical (non-Bayesian) optimal control problems.

First we state our assumptions. We assume that MDP is weakly communicating. This is a standard assumption and under this assumption, the optimal average loss satisfies the Bellman equation. Further, we assume that the dynamics are parametrized by a scalar parameter and satisfy a smoothness condition.

**Assumption A1** *(Lipschitz Dynamics)* There exist a constant $C$ such that for any state $x$ and action $a$ and parameters $\theta, \theta' \in \Theta \subseteq \Re$,

$$\|P(.|x, a, \theta) - P(.|x, a, \theta')\|_1 \leq C |\theta - \theta'| .$$

We also make a concentrating posterior assumption, which states that the variance of the difference between the true parameter and the sampled parameter gets smaller as more samples are gathered.

**Assumption A2** *(Concentrating Posterior)* Let $N_j$ be one plus the number of steps in the first $j$ episodes. Let $\widetilde{\theta}_j$ be sampled from the posterior at the current episode $j$. Then there exists a constant $C'$ such that

$$\max_j \mathbb{E}\left[N_{j-1}\left|\theta_* - \widetilde{\theta}_j\right|^2\right] \leq C' \log T .$$

The A2 assumption simply says the variance of posterior decreases given more data. In other words, we assume that the problem is learnable and not a degenerate case. A2 was actually shown to hold for two general categories of problems, finite MDPs and linearly parametrized problems with Gaussian noise Abbasi-Yadkori and Szepesvári [2015]. In addition, in this paper we prove how this assumption is satisfied for a large class of practical problems, such as smoothly parametrized sequential recommendation systems in Section 6.

Now we are ready to state the main theorem. We show a sketch of the analysis in the next section. More details are in the appendix.

**Theorem 1** *Under Assumption A1 and A2, the regret of the DS-PSRL algorithm is bounded as*

$$R_T = \widetilde{O}(C\sqrt{C'T}),$$

*where the $\widetilde{O}$ notation hides logarithmic factors.*

Notice that the regret bound in Theorem 1 does not directly depend on $S$ or $A$. Moreover, notice that the regret bound is smaller if the Lipschitz constant $C$ is smaller or the posterior concentrates faster (i.e. $\widetilde{C}'$ is smaller).

## 4 Sketch of Analysis

To analyze the algorithm shown in Figure 1, first we decompose the regret into a number of terms, which are then bounded one by one. Let $\widetilde{x}_{t+1}^a \sim P(.\,|\,x_t, a, \widetilde{\theta}_t)$, i.e. an imaginary next state sample assuming we take action $a$ in state $x_t$ when parameter is $\widetilde{\theta}_t$. Also let $\widetilde{x}_{t+1} \sim P(.\,|\,x_t, a_t, \widetilde{\theta}_t)$ and $x_{t+1} \sim P(.\,|\,x_t, a_t, \theta_*)$. By the average cost Bellman optimality equation [Bertsekas, 1995], for a system parametrized by $\widetilde{\theta}_t$, we can write

$$J(\widetilde{\theta}_t) + h_t(x_t) = \min_{a \in \mathcal{A}} \left\{ \ell(x_t, a) + \mathbb{E}\left[ h_t(\widetilde{x}_{t+1}^a) \,|\, \mathcal{F}_t, \widetilde{\theta}_t \right] \right\} . \tag{3}$$

Here $h_t(x) = h(x, \widetilde{\theta}_t)$ is the differential value function for a system with parameter $\widetilde{\theta}_t$. We assume there exists $H > 0$ such that $h_t(x) \in [0, H]$ for any $x \in \mathcal{X}$. Because the algorithm takes the optimal action with respect to parameter $\widetilde{\theta}_t$ and $a_t$ is the action at time $t$, the right-hand side of the above equation is minimized and thus

$$J(\widetilde{\theta}_t) + h_t(x_t) = \ell(x_t, a_t) + \mathbb{E}\left[ h_t(\widetilde{x}_{t+1}) \,|\, \mathcal{F}_t, \widetilde{\theta}_t \right] . \tag{4}$$

The regret decomposes into two terms as shown in Lemma 2.

**Lemma 2** *We can decompose the regret as follows:*

$$R_T = \sum_{t=1}^{T} \mathbb{E}\left[ \ell(x_t, a_t) - J(\theta_*) \right] \leq H \sum_{t=1}^{T} \mathbb{E}\left[ \mathbf{1}\{A_t\} \right] + \sum_{t=1}^{T} \mathbb{E}\left[ h_t(x_{t+1}) - h_t(\widetilde{x}_{t+1}) \right] + H$$

*where $A_t$ denotes the event that the algorithm has changed its policy at time $t$.*

The first term $H \sum_{t=1}^{T} \mathbb{E}\left[ \mathbf{1}\{A_t\} \right]$ is related to the sequential changes in the differential value functions, $h_{t+1} - h_t$. We control this term by keeping the number of switches small; $h_{t+1} = h_t$ as long as the same parameter $\widetilde{\theta}_t$ is used. Notice that under DS-PSRL, $\sum_{t=1}^{T} \mathbf{1}\{A_t\} \leq \log_2(T)$ always holds. Thus, the first term can be bounded by $H \sum_{t=1}^{T} \mathbb{E}\left[ \mathbf{1}\{A_t\} \right] \leq H \log_2(T)$.

The second term $\sum_{t=1}^{T} \mathbb{E}\left[ h_t(x_{t+1}) - h_t(\widetilde{x}_{t+1}) \right]$ is related to how fast the posterior concentrates around the true parameter vector. To simplify the exposition, we define

$$\Delta_t = \int_{\mathcal{X}} \left( P(x \,|\, x_t, a_t, \theta_*) - P(x \,|\, x_t, a_t, \widetilde{\theta}_t) \right) h_t(x) dx$$
$$= \mathbb{E}\left[ h_t(x_{t+1}) - h_t(\widetilde{x}_{t+1}) | x_t, a_t \right] .$$

Recall that $\widetilde{x}_{t+1} \sim P(.\,|\,x_t, a_t, \widetilde{\theta}_t)$ while $x_{t+1} \sim P(.\,|\,x_t, a_t, \theta_*)$, thus, from the tower rule, we have

$$\mathbb{E}\left[ \Delta_t \right] = \mathbb{E}\left[ h_t(x_{t+1}) - h_t(\widetilde{x}_{t+1}) \right] .$$

The following two lemmas bound $\sum_{t=1}^{T} \mathbb{E}\left[ \Delta_t \right]$ under Assumption A1 and A2.

**Lemma 3** *Under Assumption A1, let $m$ be the number of schedules up to time $T$, we can show:*

$$\mathbb{E}\left[ \sum_{t=1}^{T} \Delta_t \right] \leq CH \sqrt{ T\mathbb{E}\left[ \sum_{j=1}^{m} M_j \left| \theta_* - \widetilde{\theta}_j \right|^2 \right] }$$

*where $M_j$ is the number of steps in the $j$th episode.*

**Lemma 4** *Given Assumption A2 we can show:*

$$\mathbb{E}\left[ \sum_{j=1}^{m} M_j \left| \theta_* - \widetilde{\theta}_j \right|^2 \right] \leq 2C' \log^2 T .$$

Thus,

$$\mathbb{E}\left[\sum_{t=1}^{T}\Delta_t\right] \le CH\sqrt{2C'T\log^2 T} = O(\sqrt{T}\log T) \ .$$

Combining the above results, we have

$$R_T \le H\log_2(T) + CH\sqrt{2C'T\log^2 T} + H = O(CH\sqrt{C'T}\log T) \ .$$

This concludes the proof.

## 5 Experiments

In this section we compare through simulations the performance of DS-PSRL algorithm with the latest PSRL algorithm called Thompson Sampling with dynamic episodes (TSDE) Ouyang *et al.* [2017b]. We experimented with the RiverSwim environment Strehl and Littman [2008], which was the domain used to show how TSDE outperforms all known existing algorithms in Ouyang *et al.* [2017b]. The RiverSwim example models an agent swimming in a river who can choose to swim either left or right. The MDP consists of $K$ states arranged in a chain with the agent starting in the leftmost state ($s = 1$). If the agent decides to move left i.e with the river current then he is always successful but if he decides to move right he might 'fail' with some probability. The reward function is given by: $r(s, a) = 5$ if $s = 1$, $a = $ left; $r(s, a) = 10000$ if $s = K$, $a = $ right; and $r(s, a) = 0$ otherwise.

### 5.1 Scalar Parametrization

For scalar parametrization a scalar value defines the transition dynamics of the whole MDP. We did two types of experiments, In the first experiment the transition dynamics (or fail probability) were the same for all states for a given scalar value. In the second experiment we allowed for a single scalar value to define different fail probabilities for different states. We assumed two probabilities of failure, a high probability $P_1$ and a low probability $P_2$. We assumed we have two scalar values $\{\theta_1, \theta_2\}$. We compared with an algorithm that switches every time-step, which we otherwise call t-mod-1, with TSDE and DS-PSRL algorithms. We assumed the true model of the world was $\theta_* = \theta_2$ and that the agent starts in the left-most state.

In the first experiment, $\theta_1$ sets $P_1$ to be the fail probability for all states and $\theta_2$ sets $P_2$ to be the fail probability for all states. For $\theta_1$ the optimal policy was to go left for the states closer to left and right for the states closer to right. For $\theta_2$ the optimal policy was to always go right. The results are shown in Figure 2(a), where all schedules are quickly learning to optimize the reward.

In the second experiment, $\theta_1$ sets $P_1$ to be the fail probability for all states. And $\theta_2$ sets $P_1$ for the first few states on the left-end, and $P_2$ for the remaining. The optimal policies were similar to the first experiment. However the transition dynamics are the same for states closer to the left-end, while the polices are contradicting. For $\theta_1$ the optimal policy is to go left and for $\theta_2$ the optimal policy is to go right for states closer to the left-end. This leads to oscillating behavior when uncertainty about the true $\theta$ is high and policy switching is done frequently. The results are shown in Figure 2(b) where t-mod-1 and TSDE underperform significantly. Nonetheless, when the policy is switched after multiple interactions, the agent is likely to end up in parts of the space where it becomes easy to identify the true model of the world. The second experiment is an example where multi-step exploration is necessary.

### 5.2 Multiple Parameters

Even though our theoretical analysis does not account for the case with multiple parameters, we tested empirically our algorithm with multiple parameters. We assumed a Dirichlet prior for every state and action pair. The initial parameters of the priors were set to one (uniform) for the non-zero transition probabilities of the RiverSwim problem and zero otherwise. Updating the posterior in this case is equivalent to updating the parameters after every transition. We did not compare with the t-mod-1 schedule, due to the computational cost of sampling and solving an MDP every time-step. Unlike the

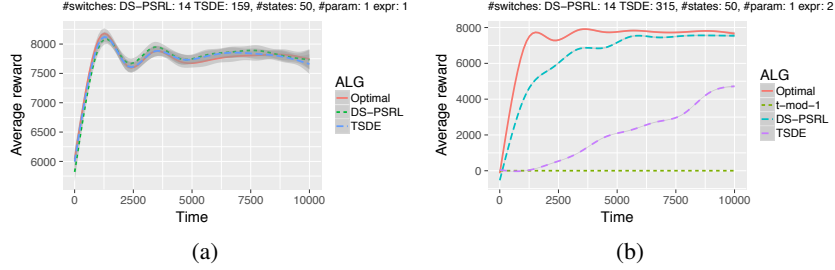

(a)　　　　　　　　　　　　　(b)

Figure 2: When multi-step exploration is necessary DS-PSRL outperforms.

scalar case we cannot define a small finite number of values, for which we can pre-compute the MDP policies. The ground truth model used was $\theta_2$ from the second scalar experiment. Our results are shown in Figures 3(a) and 3(b). DS-PSRL performs better than TSDE as we increase the number of parameters.

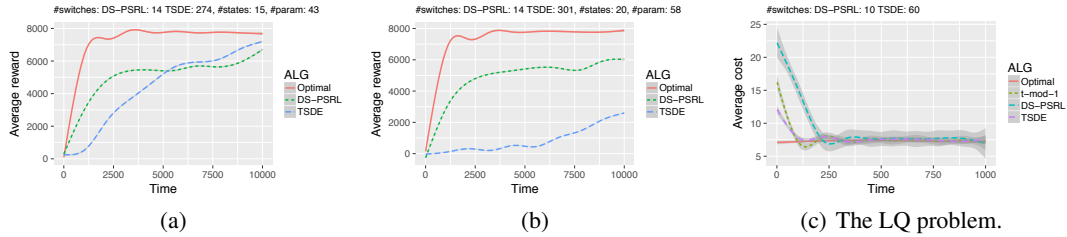

(a)　　　　　　　　(b)　　　　　　　　(c) The LQ problem.

Figure 3: Multiple parameters (a,b) and continuous domain (c).

## 5.3 Continuous Domains

In a final experiment we tested the ability of DS-PSRL algorithm in continuous state and action domains. Specifically, we implemented the discrete infinite horizon linear quadratic (LQ) problem in Abbasi-Yadkori and Szepesvári [2015, 2011]:

$$x_{t+1} = A_* x_t + B_* u_t + w_{t+1} \text{ and } c_t = x_t^T Q x_t + u_t^T R u_t,$$

where $t = 0, 1, ..., u_t \in R^d$ is the control at time $t$, $x_t \in R^n$ is the state at time $t$, $c_t \in R$ is the cost at time $t$, $w_{t+1}$ is the 'noise', $A_* \in R^{n \times n}$ and $B_* \in R^{n \times d}$ are unknown matrices while $Q \in R^{n \times n}$ and $R \in R^{d \times d}$ are known (positive definite) matrices. The problem is to design a controller based on past observations to minimize the average expected cost. Uncertainty is modeled as a multivariate normal distribution. In our experiment we set $n = 2$ and $d = 2$.

We compared DS-PSRL with t-mod-1 and a recent TSDE algorithm for learning-based control of unknown linear systems with Thompson Sampling Ouyang *et al.* [2017a]. This version of TSDE uses two dynamic conditions. The first condition is the same as in the discrete case, which activates when episodes increase by one from the previous episode. The second condition activates when the determinant of the sample covariance matrix is less than half of the previous value. All algorithms learn quickly the optimal $A_*$ and $B_*$ as shown in Figure 3(c). The fact that switching every time-step works well indicates that this problem does not require multi-step exploration.

## 6 Application to Sequential Recommendations

With 'sequential recommendations' we refer to the problem where a system recommends various 'items' to a person over time to achieve a long-term objective. One example is a recommendation system at a website that recommends various offers. Another example is a tutorial recommendation system, where the sequence of tutorials is important in advancing the user from novice to expert

over time. Finally, consider a points of interest recommendation (POI) system, where the system recommends various locations for a person to visit in a city, or attractions in a theme park. Personalized sequential recommendations are not sufficiently discussed in the literature and are practically non-existent in the industry. This is due to the increased difficulty in accurately modeling long-term user behaviors and non-myopic decision making. Part of the difficulty arises from the fact that there may not be a previous sequential recommendation system deployed for data collection, otherwise known as the cold start problem.

Fortunately, there is an abundance of sequential data in the real world. These data is usually 'passive' in that they do not include past recommendations. A practical approach that learns from passive data was proposed in Theocharous *et al.* [2017]. The idea is to first learn a model from passive data that predicts the next activity given the history of activities. This can be thought of as the 'no-recommendation' or passive model. To create actions for recommending the various activities, the authors perturb the passive model. Each perturbed model increases the probability of following the recommendations, by a different amount. This leads to a set of models, each one with a different 'propensity to listen'. In effect, they used the single 'propensity to listen' parameter to turn a passive model into a set of active models. When there are multiple model one can use online algorithms, such as posterior sampling for Reinforcement learning (PSRL) to identify the best model for a new user [Strens, 2000; Osband *et al.*, 2013]. In fact, the algorithm used in Theocharous *et al.* [2017] was a deterministic schedule PSRL algorithm. However, there was no theoretical analysis. The perturbation function used was the following:

$$P(s|X, a, \theta) = \begin{cases} P(s|X)^{1/\theta}, & \text{if } a = s \\ P(s|X)/z(\theta), & \text{otherwise} \end{cases} \tag{5}$$

where $s =$ is a POI, $X = (s_1, s_2 \ldots s_t)$ a history of POIs, and $z(\theta) = \frac{\sum_{s \neq a} P(s|X)}{1 - P(s=a|X)^{1/\theta}}$ is a normalizing factor. Here we show how this model satisfies both assumptions of our regret analysis.

**Lipschitz Dynamics**   We first prove that the dynamics are Lipschitz continuous:

**Lemma 5**  *(Lipschitz Continuity) Assume the dynamics are given by Equation 5. Then for all $\theta, \theta' \geq 1$ and all $X$ and $a$, we have*

$$\|P(\cdot|X, a, \theta) - P(\cdot|X, a, \theta')\|_1 \leq \frac{2}{e}|\theta - \theta'|.$$

Please refer to Appendix D for the proof of this lemma.

**Concentrating Posterior**   As is detailed in Appendix E (see Lemma 6), we can also show that Assumption A2 holds in this POI recommendation example. Specifically, we can show that under mild technical conditions, we have

$$\max_j \mathbb{E}\left[ N_{j-1} \left| \theta_* - \widetilde{\theta}_j \right|^2 \right] = O(1)$$

## 7   Summary and Conclusions

We proposed a practical general case PSRL algorithm, called DS-PSRL with provable guarantees. The algorithm has similar regret to state-of-the-art. However, our result is more generally applicable to continuous state-action problems; when dynamics of the system is parametrized by a scalar, our regret is independent of the number of states. In addition, our algorithm is practical. The algorithm provides for generalization, and uses a deterministic policy switching schedule of logarithmic order, which is independent from the true model of the world. This leads to efficiency in sample, space and time complexities. We demonstrated empirically how the algorithm outperforms state-of-the-art PSRL algorithms. Finally, we showed how the assumptions satisfy a sensible parametrization for a large class of problems in sequential recommendations.

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
