[Supplementary Material · psrl-nips_2018_supplementary.pdf]

# Appendices

## A  Proof of lemma 2

**Proof.** For deterministic schedule,

$$\mathbb{E}\left[J(\theta_*)\right] = \mathbb{E}\left[J(\widetilde{\theta}_t)\right] \ .$$

Thus we can write

$$
\begin{aligned}
R_T &= \sum_{t=1}^{T} \mathbb{E}\left[\ell(x_t, a_t) - J(\theta_*)\right] \\
&= \sum_{t=1}^{T} \mathbb{E}\left[\ell(x_t, a_t) - J(\widetilde{\theta}_t)\right] \\
&= \sum_{t=1}^{T} \mathbb{E}\left[h_t(x_t) - \mathbb{E}\left[h_t(\widetilde{x}_{t+1}) \mid \mathcal{F}_t, \widetilde{\theta}_t\right]\right] \\
&= \sum_{t=1}^{T} \mathbb{E}\left[h_t(x_t) - h_t(\widetilde{x}_{t+1})\right] \ .
\end{aligned}
$$

Thus, we can bound the regret using

$$
\begin{aligned}
R_T &= \mathbb{E}\left[h_1(x_1) - h_{T+1}(x_{T+1})\right] \\
&\quad + \sum_{t=1}^{T} \mathbb{E}\left[h_{t+1}(x_{t+1}) - h_t(\widetilde{x}_{t+1})\right] \\
&\leq H + \sum_{t=1}^{T} \mathbb{E}\left[h_{t+1}(x_{t+1}) - h_t(\widetilde{x}_{t+1})\right] \ ,
\end{aligned}
$$

where the second inequality follows because $h_1(x_1) \leq H$ and $-h_{T+1}(x_{T+1}) \leq 0$. Let $A_t$ denote the event that the algorithm has changed its policy at time t. We can write

$$
\begin{aligned}
R_T - H &\leq \sum_{t=1}^{T} \mathbb{E}\left[h_{t+1}(x_{t+1}) - h_t(\widetilde{x}_{t+1})\right] \\
&= \sum_{t=1}^{T} \mathbb{E}\left[h_{t+1}(x_{t+1}) - h_t(x_{t+1})\right] \\
&\quad + \sum_{t=1}^{T} \mathbb{E}\left[h_t(x_{t+1}) - h_t(\widetilde{x}_{t+1})\right] \\
&\leq H \sum_{t=1}^{T} \mathbb{E}\left[\mathbf{1}\left\{A_t\right\}\right] \\
&\quad + \sum_{t=1}^{T} \mathbb{E}\left[h_t(x_{t+1}) - h_t(\widetilde{x}_{t+1})\right] \ .
\end{aligned}
$$

∎

## B  Proof of lemma 3

**Proof.** By Cauchy-Schwarz inequality and Lipschitz dynamics assumption,

$$
\begin{aligned}
\Delta_t &\leq \left\|P(.|x_t, a_t, \theta_*) - P(.|x_t, a_t, \widetilde{\theta}_t)\right\|_1 \|h_t\|_\infty \\
&\leq CH \left|\theta_* - \widetilde{\theta}_t\right| \ .
\end{aligned}
$$

Recall that $\widetilde{\theta}_t = \widetilde{\theta}_{\tau_t}$. Let $T_j$ be the length of episode $j$. Because we have $m$ episodes, we can write

$$\sum_{t=1}^{T} \Delta_t \leq \sqrt{T \sum_{t=1}^{T} \Delta_t^2}$$

$$= CH \sqrt{T \sum_{j=1}^{m} \sum_{s=1}^{T_j} \left| \theta_* - \widetilde{\theta}_j \right|^2}$$

$$= CH \sqrt{T \sum_{j=1}^{m} M_j \left| \theta_* - \widetilde{\theta}_j \right|^2},$$

where $M_j$ is the number of steps in the $j$th episode. Thus

$$\mathbb{E}\left[ \sum_{t=1}^{T} \Delta_t \right] \leq CH \mathbb{E}\left[ \sqrt{T \sum_{j=1}^{m} M_j \left| \theta_* - \widetilde{\theta}_j \right|^2} \right]$$

$$\leq CH \sqrt{T \mathbb{E}\left[ \sum_{j=1}^{m} M_j \left| \theta_* - \widetilde{\theta}_j \right|^2 \right]}.$$

$\blacksquare$

## C  Proof of lemma 4

**Proof.** Let $S = \mathbb{E}\left[ \sum_{j=1}^{m} M_j \left| \theta_* - \widetilde{\theta}_j \right|^2 \right]$. Let $N_j$ be one plus the number of steps in the first $j$ episodes. So $N_j = N_{j-1} + M_j$ and $N_0 = 1$. We write

$$S = \mathbb{E}\left[ \sum_{j=1}^{m} N_{j-1} \left| \theta_* - \widetilde{\theta}_j \right|^2 \frac{M_j}{N_{j-1}} \right]$$

$$\overset{(a)}{\leq} 2\mathbb{E}\left[ \sum_{j=1}^{m} N_{j-1} \left| \theta_* - \widetilde{\theta}_j \right|^2 \right]$$

$$\overset{(b)}{\leq} 2 \log T \max_j \mathbb{E}\left[ N_{j-1} \left| \theta_* - \widetilde{\theta}_j \right|^2 \right]$$

$$\overset{(c)}{\leq} 2C' \log^2 T,$$

where (a) follows from the fact that $M_j / N_{j-1} \leq 2$ for all $j$, (b) follows from

$$\mathbb{E}\left[ \sum_{j=1}^{m} N_{j-1} \left| \theta_* - \widetilde{\theta}_j \right|^2 \right] \leq m \max_j \mathbb{E}\left[ N_{j-1} \left| \theta_* - \widetilde{\theta}_j \right|^2 \right]$$

and $m \leq \log T$, and (c) follows from Assumption A2.

$\blacksquare$

# D  Proof of lemma 5

**Proof.** To simplify the expositions, we use $p$ to denote $P(s = a|X)$ in this proof. Notice that $z(\theta) = \frac{1-p}{1-p^{1/\theta}}$. Based on the definition of $\|\cdot\|_1$, we have

$$\|P(\cdot|X, a, \theta) - P(\cdot|X, a, \theta')\|_1$$

$$= \left|p^{\frac{1}{\theta}} - p^{\frac{1}{\theta'}}\right| + \sum_{s \neq a}\left|\frac{P(s|X)}{z(\theta)} - \frac{P(s|X)}{z(\theta')}\right|$$

$$= \left|p^{\frac{1}{\theta}} - p^{\frac{1}{\theta'}}\right| + \left|\frac{1 - p^{1/\theta}}{1-p} - \frac{1 - p^{1/\theta'}}{1-p}\right| \sum_{s \neq a} P(s|X)$$

$$= \left|p^{\frac{1}{\theta}} - p^{\frac{1}{\theta'}}\right| + \left|\frac{1 - p^{1/\theta}}{1-p} - \frac{1 - p^{1/\theta'}}{1-p}\right| (1 - p)$$

$$= 2\left|p^{\frac{1}{\theta}} - p^{\frac{1}{\theta'}}\right|. \tag{6}$$

We also define $h(\theta, p) \triangleq p^{\frac{1}{\theta}}$. Based on calculus, we have

$$\frac{\partial h}{\partial \theta}(\theta, p) = p^{\frac{1}{\theta}} \log\left(\frac{1}{p}\right)\frac{1}{\theta^2}$$

$$\frac{\partial^2 h}{\partial \theta \partial p}(\theta, p) = \frac{1}{\theta^2} p^{\frac{1}{\theta}-1}\left[\frac{1}{\theta}\log\left(\frac{1}{p}\right) - 1\right]. \tag{7}$$

The first equation implies that $h$ is strictly increasing in $\theta$, and the second equation implies that for all $\theta > 0$, $\frac{\partial h}{\partial \theta}(\theta, p)$ is maximized by setting $p = \exp(-\theta)$. This implies that for all $\theta > 0$, we have

$$0 < \frac{\partial h}{\partial \theta}(\theta, p) \leq \frac{\partial h}{\partial \theta}(\theta, \exp(-\theta)) = \frac{1}{e\theta}.$$

Hence, for all $\theta \geq 1$, we have $0 < \frac{\partial h}{\partial \theta}(\theta, p) \leq \frac{1}{e\theta} \leq \frac{1}{e}$. Consequently, $h(\theta, p)$ as a function of $\theta$ is globally $\left(\frac{1}{e}\right)$-Lipschitz continuous for $\theta \geq 1$. So we have

$$\|P(\cdot|X, a, \theta) - P(\cdot|X, a, \theta')\|_1 = 2\left|p^{\frac{1}{\theta}} - p^{\frac{1}{\theta'}}\right| \leq \frac{2}{e}|\theta - \theta'|.$$

∎

# E  Posterior Concentration for POI Recommendation

Recall that the parameter space $\Theta = \{\theta_1, \ldots, \theta_K\}$ is a finite set, and $\theta_*$ is the true parameter. Notice that if $P(s_t = a_t|X_t)$ is close to 0 or 1, then the DS-PSRL will not learn much about $\theta_*$ at time $t$, since in such cases $P(s_t|X_t, a_t, \theta)$'s are roughly the same for all $\theta \in \Theta$. Hence, to derive the concentration result, we make the following simplifying assumption:

$$\Delta_P \leq P(s|X) \leq 1 - \Delta_P \quad \forall(X, s)$$

for some $\Delta_P \in (0, 0.5)$. Moreover, we assume that all the elements in $\Theta$ are distinct, and define

$$\Delta_\theta \triangleq \min_{\theta \in \Theta, \theta \neq \theta_*} |\theta - \theta_*|$$

as the minimum gap between $\theta_*$ and another $\theta \neq \theta_*$. To simplify the exposition, we also define

$$B \triangleq 2\max\left\{\max_{\theta \in \Theta}\max_{p \in [\Delta_P, 1-\Delta_P]}\left|\log\left(\frac{p^{1/\theta}}{p^{1/\theta_*}}\right)\right|,\right.$$

$$\left.\max_{\theta \in \Theta}\max_{p \in [\Delta_P, 1-\Delta_P]}\left|\log\left(\frac{1 - p^{1/\theta}}{1 - p^{1/\theta_*}}\right)\right|\right\}$$

$$c_0 \triangleq \frac{\min\left\{\ln\left(\frac{1}{\Delta_P}\right)\Delta_P, \ln\left(\frac{1}{1-\Delta_P}\right)(1 - \Delta_P)\right\}}{(\max_{\theta \in \Theta}\theta)^2}$$

$$\kappa \triangleq \left(\max_{\theta \in \Theta}\theta - \min_{\theta \in \Theta}\theta\right)^2.$$

Then we have the following lemma about the concentrating posterior of this problem:

**Lemma 6** *(Concentration) Assume that $\theta_t$ is sampled from $P_t$ at time step $t$, then under the above assumptions, for any $t > 2$, we have*

$$\mathbb{E}\left[(\theta_t - \theta_*)^2\right] \leq \frac{3}{ec_0^2 t} \frac{1 - P_0(\theta_*)}{P_0(\theta_*)} \times$$

$$\exp\left\{-c_0^2 \Delta_\theta^2 t + \sqrt{2B^2 t \ln(K\kappa t^2)}\right\} + \frac{1}{t^2},$$

*where $B$, $c_0$, and $\kappa$ are constants defined above. Note that they only depend on $\Delta_P$ and $\Theta$*

Notice that Lemma 6 implies that

$$t\mathbb{E}\left[(\theta_t - \theta_*)^2\right] \leq O\left(\exp\left\{-c_0^2 \Delta_\theta^2 t + \sqrt{2B^2 t \ln(K\kappa t^2)}\right\}\right) + \frac{1}{t} = O(1)$$

for any $t > 2$. This directly implies that $\max_j \mathbb{E}\left[N_{j-1} \left|\theta_* - \widetilde{\theta}_j\right|^2\right] = O(1)$. Q.E.D.

### E.1 Proof of lemma 6

**Proof.** We use $P_0$ to denote the prior over $\theta$, and use $P_t$ to denote the posterior distribution over $\theta$ at the end of time $t$. Note that by Bayes rule, we have

$$P_t(\theta) \propto P_0(\theta) \prod_{\tau=1}^t P(s_\tau | X_\tau, a_\tau, \theta) \quad \forall t \text{ and } \forall \theta \in \Theta.$$

We also define the posterior log-likelihood of $\theta$ at time $t$ as

$$\Lambda_t(\theta) = \log\left\{\frac{P_t(\theta)}{P_t(\theta_*)}\right\} = \log\left\{\frac{P_0(\theta)}{P_0(\theta_*)} \prod_{\tau=1}^t \left[\frac{P(s_\tau | X_\tau, a_\tau, \theta)}{P(s_\tau | X_\tau, a_\tau, \theta_*)}\right]\right\}$$

for all $t$ and all $\theta \in \Theta$. Notice that $P_t(\theta) \leq \exp\left[\Lambda_t(\theta)\right]$ always holds, and $\Lambda_t(\theta_*) = 0$ by definition. We also define $p_t \triangleq P(s_t = a_t | X_t)$ to simplify the exposition. Note that by definition, we have

$$P(s_t | X_t, a_t, \theta) = \begin{cases} p_t^{1/\theta} & \text{if } s_t = a_t \\ \frac{P(s_t | X_t)}{1 - p_t}(1 - p_t^{1/\theta}) & \text{otherwise} \end{cases}$$

Define the indicator $z_t = \mathbf{1}\{s_t = a_t\}$, then we have

$$\log\left\{\frac{P(s_t | X_t, a_t, \theta)}{P(s_t | X_t, a_t, \theta_*)}\right\} = z_t \log\left[\frac{p_t^{1/\theta}}{p_t^{1/\theta_*}}\right] + (1 - z_t) \log\left[\frac{1 - p_t^{1/\theta}}{1 - p_t^{1/\theta_*}}\right]$$

Since $p_t$ is $\mathcal{F}_{t-1}$-adaptive, we have

$$\mathbb{E}\left[\log\left\{\frac{P(s_t | X_t, a_t, \theta)}{P(s_t | X_t, a_t, \theta_*)}\right\}\Big| \mathcal{F}_{t-1}, \theta_*\right]$$

$$= p_t^{1/\theta_*} \log\left[\frac{p_t^{1/\theta}}{p_t^{1/\theta_*}}\right] + (1 - p_t^{1/\theta_*}) \log\left[\frac{1 - p_t^{1/\theta}}{1 - p_t^{1/\theta_*}}\right]$$

$$= -\mathrm{D}_{\mathrm{KL}}\left(p_t^{1/\theta_*} \| p_t^{1/\theta}\right) \leq -2\left(p_t^{1/\theta_*} - p_t^{1/\theta}\right)^2,$$

where the last inequality follows from Pinsker's inequality. Notice that function $h(x) = p_t^x$ is a strictly convex function of $x$, and $\frac{dh}{dx}(x) = p_t^x \ln(p_t)$, we have

$$p_t^{1/\theta} - p_t^{1/\theta_*} \geq \ln(p_t) p_t^{1/\theta_*}(1/\theta - 1/\theta_*) = \ln(1/p_t) p_t^{1/\theta_*} \frac{(\theta - \theta_*)}{\theta \theta_*}$$

Similarly, we have $p_t^{1/\theta_*} - p_t^{1/\theta} \geq \ln(1/p_t) p_t^{1/\theta} \frac{(\theta_* - \theta)}{\theta \theta_*}$. Consequently, we have

$$\left|p_t^{1/\theta} - p_t^{1/\theta_*}\right| \geq \ln(1/p_t) \min\left\{p_t^{1/\theta_*}, p_t^{1/\theta}\right\} \frac{|\theta - \theta_*|}{\theta \theta_*}$$

$$\geq \ln(1/p_t) p_t \frac{|\theta - \theta_*|}{\theta \theta_*},$$

where the last inequality follows from the fact $\theta, \theta_* \in [1, \infty)$. Since function $\ln(1/x)x$ is concave on $[0, 1]$ and $p_t \in [\Delta_P, 1 - \Delta_P]$, we have $\ln(1/p_t)p_t \geq \min\{\ln(1/\Delta_P)\Delta_P, \ln(1/(1 - \Delta_P))(1 - \Delta_P)\}$. Define

$$c_0 \triangleq \frac{\min\{\ln(1/\Delta_P)\Delta_P, \ln(1/(1 - \Delta_P))(1 - \Delta_P)\}}{(\max_{\theta \in \Theta} \theta)^2}, \tag{8}$$

then we have $\left| p_t^{1/\theta} - p_t^{1/\theta_*} \right| \geq c_0 |\theta - \theta_*|$. Hence we have

$$-\mathrm{D_{KL}}\left(p_t^{1/\theta_*} \| p_t^{1/\theta}\right) \leq -2c_0^2(\theta - \theta_*)^2.$$

Furthermore, we define

$$\xi_t(\theta) \triangleq \log\left\{\frac{P(s_t|X_t, a_t, \theta)}{P(s_t|X_t, a_t, \theta_*)}\right\}$$
$$- \mathbb{E}\left[\log\left\{\frac{P(s_t|X_t, a_t, \theta)}{P(s_t|X_t, a_t, \theta_*)}\right\} \bigg| \mathcal{F}_{t-1}, \theta_*\right]. \tag{9}$$

Obviously, by definition, $\mathbb{E}\left[\xi_t(\theta)|\mathcal{F}_{t-1}, \theta_*\right] = 0$. We also define

$$B \triangleq 2\max\left\{\max_{\theta \in \Theta} \max_{p \in [\Delta_P, 1 - \Delta_P]} \left|\log\left(\frac{p^{1/\theta}}{p^{1/\theta_*}}\right)\right|,\right.$$
$$\left.\max_{\theta \in \Theta} \max_{p \in [\Delta_P, 1 - \Delta_P]} \left|\log\left(\frac{1 - p^{1/\theta}}{1 - p^{1/\theta_*}}\right)\right|\right\}, \tag{10}$$

then $|\xi_t(\theta)| \leq B$ always holds. This allows us to use Azuma's inequality. Specifically, for any $\theta \in \Theta$, any $t$, and any $\delta \in (0, 1)$, we have $\sum_{\tau=1}^{t} \xi_\tau(\theta) \leq \sqrt{2B^2 t \ln(K/\delta)}$ with probability at least $1 - \delta/K$. Taking a union bound over $\theta \in \Theta$, we have

$$\sum_{\tau=1}^{t} \xi_\tau(\theta) \leq \sqrt{2B^2 t \ln(K/\delta)} \quad \forall \theta \in \Theta \tag{11}$$

with probability at least $1 - \delta$. Consequently, we have

$$\Lambda_t(\theta) = \log\left\{\frac{P_0(\theta)}{P_0(\theta_*)}\right\}$$
$$+ \sum_{\tau=1}^{t}\left\{z_\tau \log\left[\frac{p_\tau^{1/\theta}}{p_\tau^{1/\theta_*}}\right] + (1 - z_\tau)\log\left[\frac{1 - p_\tau^{1/\theta}}{1 - p_\tau^{1/\theta_*}}\right]\right\}$$
$$= \log\left\{\frac{P_0(\theta)}{P_0(\theta_*)}\right\} - \sum_{\tau=1}^{t}\mathrm{D_{KL}}\left(p_\tau^{1/\theta_*} \| p_\tau^{1/\theta}\right) + \sum_{\tau=1}^{t}\xi_\tau(\theta)$$
$$\leq \log\left\{\frac{P_0(\theta)}{P_0(\theta_*)}\right\} - 2c_0^2(\theta - \theta_*)^2 t + \sum_{\tau=1}^{t}\xi_\tau(\theta) \tag{12}$$

Combining the above inequality with equation 11, we have

$$\Lambda_t(\theta) \leq \log\left\{\frac{P_0(\theta)}{P_0(\theta_*)}\right\} - 2c_0^2(\theta - \theta_*)^2 t + \sqrt{2B^2 t \ln(K/\delta)} \quad \forall \theta \in \Theta$$

with probability at least $1 - \delta$. Hence, we have

$$P_t(\theta) \leq \exp\left[\Lambda_t(\theta)\right] \tag{13}$$
$$\leq \frac{P_0(\theta)}{P_0(\theta_*)}\exp\left\{-2c_0^2(\theta - \theta_*)^2 t + \sqrt{2B^2 t \ln(K/\delta)}\right\}$$

for all $\theta \in \Theta$ with probability at least $1 - \delta$. Thus, for any $\mathcal{F}_{t-1}$ s.t. the above inequality holds, we have

$$\mathbb{E}\left[(\theta_t - \theta_*)^2|\mathcal{F}_{t-1}, \theta_*\right] = \sum_{\theta \neq \theta_*} P_t(\theta)(\theta - \theta_*)^2$$
$$\leq \sum_{\theta \neq \theta_*} \frac{P_0(\theta)}{P_0(\theta_*)}\exp\left\{-2c_0^2(\theta - \theta_*)^2(t - 1)\right.$$
$$\left. + \sqrt{2B^2(t - 1)\ln(K/\delta)}\right\}(\theta - \theta_*)^2 \tag{14}$$

For $t > 2$, we have

$$\exp\left\{-c_0^2(\theta - \theta_*)^2(t-2)\right\}(\theta - \theta_*)^2 \leq \frac{1}{ec_0^2(t-2)} \leq \frac{3}{ec_0^2 t},$$

where the last inequality follows from the fact that $t - 2 \geq \frac{t}{3}$. Hence we have

$$\mathbb{E}\left[(\theta_t - \theta_*)^2 \big| \mathcal{F}_{t-1}, \theta_*\right]$$

$$\leq \frac{3}{ec_0^2 t} \sum_{\theta \neq \theta_*} \frac{P_0(\theta)}{P_0(\theta_*)} \exp\left\{-c_0^2(\theta - \theta_*)^2 t + \sqrt{2B^2 t \ln(K/\delta)}\right\}$$

$$\leq \frac{3}{ec_0^2 t} \sum_{\theta \neq \theta_*} \frac{P_0(\theta)}{P_0(\theta_*)} \exp\left\{-c_0^2 \Delta_\theta^2 t + \sqrt{2B^2 t \ln(K/\delta)}\right\}$$

$$= \frac{3}{ec_0^2 t} \frac{1 - P_0(\theta_*)}{P_0(\theta_*)} \exp\left\{-c_0^2 \Delta_\theta^2 t + \sqrt{2B^2 t \ln(K/\delta)}\right\},$$

where the second inequality follows from $(\theta - \theta_*)^2 \geq \Delta_\theta^2$. For $\mathcal{F}_{t-1}$ s.t. inequality 13 does not hold, we use the naive bound

$$(\theta_t - \theta_*)^2 \leq \kappa \triangleq \left(\max_{\theta \in \Theta} \theta - \min_{\theta \in \Theta} \theta\right)^2.$$

Since inequality 13 holds with probability at least $1 - \delta$, we have

$$\mathbb{E}\left[(\theta_t - \theta_*)^2 \big| \theta_*\right] \tag{15}$$

$$\leq \frac{3}{ec_0^2 t} \frac{1 - P_0(\theta_*)}{P_0(\theta_*)} \exp\left\{-c_0^2 \Delta_\theta^2 t + \sqrt{2B^2 t \ln(K/\delta)}\right\} + \delta\kappa.$$

Finally, by choosing $\delta = \frac{1}{\kappa t^2}$ and taking an expectation over $\theta_*$, we have

$$\mathbb{E}\left[(\theta_t - \theta_*)^2\right] \tag{16}$$

$$\leq \frac{3}{ec_0^2 t} \frac{1 - P_0(\theta_*)}{P_0(\theta_*)} \exp\left\{-c_0^2 \Delta_\theta^2 t + \sqrt{2B^2 t \ln(K\kappa t^2)}\right\} + \frac{1}{t^2}.$$

∎