[Reviews · NeurIPS 2018]

Reviewer 1



This paper scales up the previous posterior sampling RL methods, such as Thompson sampling, by extending to continuous state and action space. A Bayesian-style regret bound is given, and the author shows theoretically that the proposed algorithm has the state-of-the-art efficiency in time and space, and enjoys the applicability to continuous state and action problems. Given the merits, the paper seems to be written in a hasty manner, which makes the reading experience tougher than I expect. There are some technical concerns 1. Assumption 2 seems to be restrictive to Gaussian noise. If the assumption is violated for non-Gaussian noise, what will happen and how robust is the algorithm? 2. One major concern is that the author claims that the proposed algorithm can scale up to continuous state/action space problems. However, in the experiments, there are only discrete action domains, which makes the comparison study less convincing. 3. The authors state that result is more generally applicable to continuous state-action problems; when dynamics of the system is parameterized by a scalar, their regret scales with the square root of the number of parameters and is independent of the number of states. This is opposite to the general sense. Since regret usually increases with the number of states. This paper assumes the MDP model is parameterized by a scalar, the parameterization of MDP is related with the number states. the regret is proportional with the complexity of the model parameterization. 4. Other methods for comparison study. The paper compares through simulations the performance of DS-PSRL algorithm with the latest PSRL algorithm called Thompson Sampling with dynamic episodes (TSDE). Some other algorithms should be compared with the proposed algorithm. Some minor suggestions: pp2. The introduction is too long. A ‘Related work’ section should be split to formulate a separate section. line 169, ‘differential’ —> ‘differentiable’ line 192-194 the reference format of many places is incorrect. such as this place: there should be brackets of [Ouyang et al.] line 240-241 the TSDE work by Ouyang et al. should be introduced as early as the ‘Related work’ part. line 286: it should be “The algorithm provides generalization.”

Reviewer 2



This paper proposes a posterior sampling algorithm for reinforcement learning in non-episodic MDPs. This work can handle continuous state and action spaces but assumes that the MDP is known up to a parameter that smoothly controls the transition probabilities. O(sqrt(T)lnT) Bayesian regret bounds are proved under relatively mild assumptions and the paper shows that these assumptions are satisfied in a class of sequential recommendation problems. I like several things about this paper: - The paper is well written and easy to follow, including the proofs in the appendix. - It gives a good summary of relevant existing work and relates them well to this submission. - The simulations nicely demonstrate the practical performance of this method in various settings, though including lazyPSRL as a baseline would have been nice. - The proofs are simple but fairly novel and contain some nice insights that I have not seen elsewhere. However, I also see the following issues: - I am concerned about the significance of this work. The main purpose of this paper is to go beyond finite MDPs and similar classes but at the price of a scalar parameterization assumption. Even more severe, the algorithm has to know the dependency of the transition kernel on the scalar parameter. This is quite a restrictive assumption and highly limits the scope of this work. While it can be okay to make strong assumptions, I believe they need to be discussed and motivated properly which isn't done here (see points below). - While the paper shows that the algorithm's assumptions are satisfied in sequential recommendation problems, it does not empirically test the method in such problems. That is a missed chance to demonstrate the practical relevance of the algorithm. - From what I see in the proofs in the appendix, the scalar assumption is made to avoid a mismatch of norms of the concentrating posterior assumption and the application of Hoelder's inequality. I think the paper should discuss why the scalar is made and also discuss how one might generalize this result to the non-scalar case with potentially different assumptions.

Reviewer 3



The authors introduce DS-PSRL for reinforcement learning in 1-dimensional continuously parametrized MDPs. The algorithm is computationally efficient because the number of policy switches is fewer than O(log T) (where the major computational cost of a policy switch is solving an MDP). The authors provide a Bayesian regret bound that characterizes how the algorithm scales with the smoothness of the parametrization and how quickly the posterior distribution concentrates. The experiments suggest that DS-PSRL competes favorably against TSDE and works well even in the case where the task is parametrized by multiple parameters. The paper is clearly written and provides both theoretical insights and strong theoretical results. Post-Author-Response: I've read the response and I believe this is paper should be accepted. I view the scalar parametrization as a starting point that future work can extend. I would prefer that the authors compare to performance against multiple algorithms.